# Standard Analytical Methods, Sensory Evaluation, NIRS and Electronic Tongue for Sensing Taste Attributes of Different Melon Varieties

**DOI:** 10.3390/s19225010

**Published:** 2019-11-16

**Authors:** Dzsenifer Németh, Gábor Balázs, Hussein G. Daood, Zoltán Kovács, Zsanett Bodor, John-Lewis Zinia Zaukuu, Viktor Szentpéteri, Zoltán Kókai, Noémi Kappel

**Affiliations:** 1Department of Vegetable and Mushroom Growing, Faculty of Horticultural Science, Szent István University, Villányi út 29–43., H-1118 Budapest, Hungary; Nemeth.Dzsenifer@kertk.szie.hu (D.N.); Balazs.Gabor@kertk.szie.hu (G.B.); 2Regional Knowledge Center, Szent István University, Páter Károly utca 1., H-2100 Gödöllő, Hungary; Daood.Hussein@fh.szie.hu; 3Department of Physics and Control, Faculty of Food Science, Szent István University, Somlói út 14–16., H-1118 Budapest, Hungary; Kovacs.Zoltan3@etk.szie.hu (Z.K.); arscube@gmail.com (Z.B.); zaukuu.john-lewis.zinia@hallgato.uni-szie.hu (J.-L.Z.Z.); 4Institute of Genetics, Microbiology and Biotechnology, Department of Microbiology and Environmental Toxicology, Szent István University, Páter Károly út. 1., 2100 Gödöllő, Hungary; Szentpeteri.Viktor@hallgato.uni-szie.hu; 5Department of Postharvest Science and Sensory Evaluation, Faculty of Food Science, Szent István University, Villányi út 35-43., H-1118 Budapest, Hungary; Kokai.Zoltan@etk.szie.hu

**Keywords:** melon varieties, sensory evaluation, NIRS, electronic tongue, refraction, vitamin C content, carotenoid content

## Abstract

Grafting by vegetables is a practice with many benefits, but also with some unknown influences on the chemical composition of the fruits. Our goal was to assess the effects of grafting and storage on the extracted juice of four orange-fleshed Cantaloupe type (*Celestial*, *Donatello*, *Centro*, *Jannet*) melons and two green-fleshed Galia types (*Aikido*, *London*), using sensory profile analysis and analytical instruments: An electronic tongue (E-tongue) and near-infrared spectroscopy (NIRS). Both instruments are known for rapid qualitative and quantitative food analysis. Linear discriminant analysis (LDA) was used to classify melons according to their varieties and storage conditions. Partial least square regression (PLSR) was used to predict sensory and standard analytical parameters. *Celestial* variety had the highest intensity for sensory attributes in Cantaloupe variety. Both green and orange-fleshed melons were discriminated and predicted in LDA with high accuracies (100%) using the E-tongue and NIRS. Galia and Cantaloupe inter-varietal classification with the E-tongue was 89.9% and 82.33%, respectively. NIRS inter-varietal classification was 100% with *Celestial* variety being the most discriminated as with the sensory results. Both instruments, classified different storage conditions of melons (grafted and self-rooted) with high accuracies. PLSR showed high accuracy for some standard analytical parameters, where significant differences were found comparing different varieties in ANOVA.

## 1. Introduction

Melon is grown on 3.5–3.7 million hectares around the world. Nowadays, a strong increase in production volumes and consumption can be observed in developing countries with high populations [1]. Grafting of vegetable plants is already common practice in the word. The main argument for melon grafting is the gained increase in tolerance against pests [2], and abiotic stresses [3] and the increase in yield [4]. Several research groups also agreed that the physical characteristics of the *Solanaceae* and *Cucurbitaceae* families were not affected by grafting [3]. Fruit quality of melon is made up of several factors, like the visual appearance, texture, and the taste and aroma, which are the most important characteristics for consumers [5]. There are studies showing that grafting can influence the soluble sugar content of melon in a negative way. At the late 1940s [6] it was recorded that the *Cucurbita moschata* rootstock causes weaker texture and aroma in the grafted Honey Dew’ fruits, although it also established tolerance against fusarium wilt. Grafting did reduce sugar content by 1 Brix° in case of watermelon and melon as well in the experiments of international studies [7,8,9]. Compared to 350 melon varieties and found that the vitamin C content of them is between 0.75–35.3 mg/100g. According to Bíró and Lindner (1988), orange-fleshed melons contain 35 mg/100g vitamin C, whereas, green-fleshed melons only contain 25 mg/100g vitamin C [10]. Condurso et al. (2012) concluded that the carotenoid content of the ‘*Proteo*’ melon variety (*Cucumis melo var. reticulatus*) increased when grafted on hybrid pumpkin rootstock [11]. The experiment of Zhou et al. (2014) revealed that plants grafted to ‘*Elis*’, ‘*P360*’, ‘*RS841*’ and ‘*AS10*’ rootstocks lutein content is significantly higher, β-carotene level increases eight fold and α-carotene level increases by 56% when compared to control plants grafted on ‘*P360*’ [12]. In the study of Zhou et al. (2014) the combination of ‘*Lyu*’–‘*Nanzhen No. 1*’ plants resulted in four times higher fruit β-carotene content compared to fruits grown on nongrafted plants [12]. However, not all grafting combination is favorable on carotenoid content, which was shown in the study of Verzera et al. (2014) where the use of ‘*Energy*’ and ‘*Sting*’ rootstocks resulted in 55% decrease in β-carotene content compared to control [13]. Storage of vegetables and fruits is necessary to lengthen the consumption and processing season of them. To avoid over-ripening, it is advised to store melon at low temperature, but since it is a cold-sensitive fruit, after harvest, a gradual approach has to be followed during the cooling process until the reach of 6–10 °C which has the desired preservative effect. At 0 Celsius degree freeze damage is caused to the fruit, due to this, it becomes vitrificated, and whole units become spoilt that way [14]. The data suggest that the effects of rootstocks on flesh firmness varied depending on the rootstock and the scion [15].

Sensory properties of fruits are very important factors in consumer acceptance; therefore, objective tests for determining the preferred sensory attributes is essential. Sensory test based on ISO standards can be performed by panels for the description of different fruits, such as melon [16,17]. Electronic sensory instruments like electronic tongues and electronic noses are also available and used in food science. Electronic tongues are designed to work like human tongues, but with a higher sensitivity to flavors and aromas working like a fingerprint method. Electronic tongues were applied in several fields of food studies, for instance, classification of different varieties of fruits and vegetables, testing the authenticity of foods and beverages [18,19] or predicting sweetness and sugar content [20]. Hungarian researchers could discriminate regions of watermelon samples based on the results of the electronic tongue [21]. The electronic tongue was also used for discriminate between different processes [22] or storage conditions [18].

Also, in the scope of advanced analytical instruments for melon quality, is the near-infrared spectroscopy (NIRS). NIRS is a well-established technique that operates within a wavelength range of 700–2500 nm [23]. The NIRS principle basically encompasses the emission, absorption and reflection of light which, are dependent on the chemical composition of the product (microstructure) and its light-scattering properties. Advanced multivariate statistical techniques, such as least discriminant analysis and partial least squares regression, are then applied to extract the required information from the usually convoluted spectra [24]. Advantages of NIRS include: Waste-free rapid analysis, low cost per evaluation, and simultaneously testing for diverse properties from a single spectrum. NIRS has been used to predict the pulp color difference of melon [25] and also the sugariness and hardness of different melon varieties [26]. The principles and advantages of NIRS makes it suitable for breeding, research that often involves genotypic or phenotypic discrimination, but there is a paucity of a report about its application in this context. Considering that melons are a very important and popular fruit substitute vegetable, little scientific results have been published so far on the correlation between instrumental and organoleptic properties of melon, particularly, the influence of technology effects, such as storage or grafting.

The aim of our study was to determine the compositional differences between five melon varieties grafted and stored under different conditions (2 °C and 17 °C) and to classify the predict their compositional qualities using advanced methods (e-tongue and NIRS).

## 2. Materials and Methods

Examinations were performed from late summer to autumn in 2018. Experimental samples were gathered from several farmers. Altogether six melon varieties were examined, four orange-fleshed Cantaloupe type (*Celestial*, *Donatello*, *Centro*, *Jannet*) and two green-fleshed Galia type (*Aikido*, *London*). Two experiments were carried out, the first aimed towards comparing varieties and the second was to examine the effects of different storage conditions. Variety comparison was performed with *Celestial*, *Donatello*, *Centro*, *Aikido* and *London* varieties. Five different plants of each variety were harvested and later on used in the experiments (variety dataset) resulting in a total of 25 samples (5 varieties and five repeats each). In case of the *Jannet* variety (Cantaloupe type) we had fruits originating from grafted and self-rooted plants as well. Five different plants of each variety wereharvested and used for the storage experiments (storage dataset). The samples included fruits processed freshly and stored for 7 days at 2 °C and 17 °C were compared. 

### 2.1. Standard Analytical Methods

The use of standard analytical methods was necessary to support the data obtained by NIRS and electronic tongue measurements. The water-soluble solids content was measured with a digital hand refractometer (PAL-1, ATAGO). After preparation of the samples, a few drops of fruit juice were dropped onto the surface of the prism, and the instrument read the measurement results. The machine was then calibrated with distilled water. Brix° corresponds to the sugar percentage in the solution, i.e., 1% is equal to 1 g sugar per 100 g solution. To determine the carotenoid profile and the vitamin C content, we used the Hitachi Chromaster HPLC. The isolation and data processing were operated by EZChrom Elite software. To measure the vitamin content, samples have been injected on a reverse-phase C18 Nautilus, 150 × 4.6 mm (Macherey-Nagel, Düren, Germany) column. To determine the exact amount, a known amount of standard material (Sigma Aldrich, Budapest) was injected, and the concentration values for the peaks obtained were calculated. The maximum absorption of vitamin C was detected at 244 nm. To measure the carotenoids, after draining solvents were removed by vacuum distillation (RVO 400, Vacuubrand, Germany) at 40 °C. Samples have been filtrated through a 0.45 µm diameter PTFE HPLC filter, before injection on the column. Carotenoids were separated on a Purospher^®^ STAR RP C18 end-capped 3 µm, 250 × 4.6 mm column with 50 min long gradient elution according to the method of Daood et al. (2014) [27]. Peaks were identified by comparing their retention time and maximum absorbance to standards (Sigma Aldrich, Budapest). The carotenoids were detected between 195 and 700 nm. 

### 2.2. Sensory Tests

Samples were prepared 30 min before tasting. First, the sensory attributes and their corresponding reference values were determined, in order to reduce the variation in the resulting dataset. Then, sensory tests were performed according to ISO 13299 standard by 10 panelists who evaluated the followings: The presence of fermented aroma (since it was a storage test), sweet aroma, flesh color, texture, juiciness, sweet taste, fermented flavor, after taste, and taste persistence [28]. The combined results of the properties were plotted on profile diagrams, which were prepared by ProfiSens, a sensory analysis software. Tests were performed according to ISO 8589, and differences between data were evaluated with univariate ANOVA and Fisher LSD significance level evaluation procedures [29].

### 2.3. Electronic Tongue Measurements

Electronic tongue measurements were carried out with Alpha Astree [30] potentiometric electronic tongue, containing seven sensors (ISFET—BB, CA, HA, JB, JE, GA, ZZ), that have been developed for liquid food applications. Five tubes of each melon sample were frozen, then melted up and filtered. For electronic tongue measurements, 10 times dilution (10 mL of filled up to volume in 100 mL volumetric flask) was prepared from each sample in three replicates resulting in 15 replicate sample per type and storage level. Melons for variety and storage discrimination was measured on two following days (first day for variety test, and the second day for storage test). Each day three sequences were formed, containing two replicates for each sample. Five replicate samples were obtained from five individual plants for each group of the variety and storage data sets, respectively. Each sample was measured four times with the electronic tongue resulting in 100 readings for each variety and storage data set. The last 10 s of the sensor signals of each sample, were averaged and exported into an excel sheet for statistical analysis Figure 1. 

### 2.4. Near-Infrared Spectroscopy (NIRS) Measurements

Metri analyzer (benchtop spectrometer) was used for spectra collection. Transflectance spectra were collected using a cuvette providing 0.4 mm layer thickness of the tested melon varieties. The samples were prepared the same way as for electronic tongue measurement, but there was no dilution applied. Three consecutive scans of all the five repeats were collected, while using purified water (MQ) as a control in a spectral step of 3 nm. The spectral acquisition was performed at room temperature. In total, 75 spectra of melon samples were recorded for each variety dataset and storage dataset. Data analysis was done at the spectral range of 950–1650 nm after raw spectra assessment. Spectra pretreated was done first with detrended and Savitzky-Golay smoothing filter using second order polynomial in R-studio using the “aquap2” package [31].

### 2.5. Statistical Evaluation

Statistical evaluation of the results of standard analytical methods properties, such as Brix°, vitamin C and carotenoid profile was performed with descriptive statistics (mean and standard deviation) and ANOVA test followed by Tukey-HSD pair wised comparison between varieties and between storage types for each parameter at *p* < 0.05 significance level. Results of the electronic tongue were evaluated after drift correction of the raw results to exclude the effect of the ageing of the organic membranes, and these drift corrected results were used for the representation of the results of the ET. Results of ET and NIRS were evaluated with chemometric methods: Principal component analysis (PCA), linear discriminant analysis (LDA) and partial least square regression (PLSR). PCA was used as an exploratory analysis for identification of the outliers and describing the main patterns of the variety and storage data set. LDA was used for building classification models for different varieties and a separate model for storage conditions. Models were validated with threefold cross validation. Partial least square regression was carried out to predict the results of standard analytical methods and sensory test with leave one out cross validation. Root mean square error was also calculated for the training (RMSEC) and validation (RMSECV) data sets separately for the data set of varieties and storage measurements. Microsoft Excel, SPSS 25 and R-project 3.5.2 software were used for the data evaluation.

## 3. Results

### 3.1. Results of the Standard Analytical Measurment

#### 3.1.1. Results of the Variety Data Set

Results of standard analytical methods can be seen in Table 1. Results of ANOVA test showed a significant difference in each parameter between Cantaloupe and Galia type melons. In Galia type melons significant difference was found in violaxanthin and chlorophyll components between *Aikido* and *London* varieties. Results of Cantaloupe type melons showed a significant difference in β-carotene between *Celestial*, *Centro* and *Donatello*. Moreover, a significant difference was obtained in violaxanthin between *Centro* and *Donatello*, and in total carotene *Centro* also distinguished from the two other types and in vitamin C from *Celestial* variety.

#### 3.1.2. Results of Storage Data Set

ANOVA results showed a significantly higher carotenoid content and Brix° values for self-rooted fresh melons (Table 2). Grafted fresh melon had also higher β-carotene, mutatoxanthin, lutein and total carotene content than the stored group of melons. Highest vitamin C content was obtained for grafted melons stored at 2 °C, with significantly higher values than the other groups. However, self-rooted melons stored at 2 °C showed also significantly higher vitamin C content than the ones stored at 17 °C. Grafted melons showed significantly lower Brix° compared to self-rooted melons.

### 3.2. Results of the Classical Sensory Test

Results of variety data, (Figure 2a,b) set for the Cantaloupe type melons showed significant differences in four parameters, based on the results of ANOVA and pair wised comparison at *p* < 0.05 significance level. *Celestial* showed significantly higher fermented aroma compared to *Centro*, however, significantly higher fermented taste compared to the other two groups. Texture value of *Donatello* was significantly higher compared to *Centro* and *Celestial*, while *Centro* showed significantly lower juiciness compared to the two other variety. Comparing the Galia type melons significant difference can be observed in three parameters: *Aikido* had significantly intense fermented taste, aroma and flesh color compared to *London*.

In the case of the storage data set (Figure 2c,d) for fresh melons, significant differences between grafted and self-rooted melon were obtained for eight parameters. Fermented taste and aroma did not show a significant difference between the two before mentioned types, while with the exception of juiciness self-rooted melon showed significantly higher values compared to grafted melon. Melons stored at 2 °C showed similar results, apart from in this case there were no significant difference in flesh color and fermented flavor, while comparing the fresh ones, here grafted melon showed significantly more intense fermented aroma comparing the self-rooted melon.

### 3.3. Results of Classification Models for Electronic Tongue and NIRS Measurements

#### 3.3.1. Results of the Variety Test Set

PCA results of the electronic tongue showed a separation tendency mainly between Cantaloupe and Galia type melons based on PC1. LDA classification results showed similar results and trends (Figure 3). Detailed results of the classification model can be found in Table 3. LDA model built for the classification of the five varieties presented average recognition and prediction abilities of 85.51% and 59.01%, respectively, for the electronic tongue. *Centro* was classified correctly during training, while in validation misclassification was found belonging to *Donatello* type in 4.95%. *Aikido* and *London* varieties completely were distinguished from *Celestial*, *Donatello* and *Centro* types. This tendency can be seen on Figure 3. and have been proven by the model built for discriminating the yellow (Cantaloupe type) and green fleshed (Galia type) melons, where 100% correct classification was obtained both for recognition and prediction. 

Classification of 100% was obtained for recognition with the NIRS, but with a prediction accuracy of 54.75% for the classification of the five tested variates. *Celestial* and *Donatello* varieties were the most accurately predicted with an accuracy of 80.16% and 73.4%, respectively. During cross validation *Aikido* had the lowest prediction accuracy of 20%, and misclassification of 40%, 20%, 13.4% and 6.6% as *Donatello*, *Centro*, *London* and *Celestial* varieties, respectively. *Donatello*, *Aikido* and *London* varieties were in certain cases (6.1%) misclassified as *Celestial* variety Classification tables were also, independently built for the Galia (Table 4) and Cantaloupe (Table 5) types with the electronic tongue and NIRS.

LDA classification model built for Galia type melons provided average recognition and prediction abilities of 89.99% and 87.49%, respectively, for the electronic tongue. Recognition was 100% with 90% prediction for all Galia varieties using the NIRS. *London* variety—14.99% and 20% were misclassified as *Aikido* variety with the electronic tongue and NIRS, respectively (Table 4).

Cantaloupe type melons model resulted in average recognition and prediction abilities of 82.33% and 51.19% using data from the electronic tongue. *Centro* was classified correctly, while *Celestial* and *Donatello* types showed misclassifications belonging to each other. There was 100% recognition for all varieties and 64.47% average prediction with the NIRS. *Celestial* variety had the highest prediction accuracy of 93.4% with 6.6% being misclassified as *Donatello*. *Donatello* variety had the lowest prediction accuracy, with 40% being, misclassified as *Centro* (Table 5).

#### 3.3.2. Results of Storage Test Set

Results of PCA model for the storage test set showed the separation of the grafted and self-rooted melons through PC1, which describes the 99.01% of the total variance. Through PC2, the separation tendency of the different storage conditions was observed. LDA model built for the classification of five storage levels provided average recognition and prediction abilities of 92.01% and 87.03%, respectively. Self-rooted fresh and at 2 °C stored melons were classified correctly during training, in validation the classification accuracy was 93.4%. Grafted fresh melons showed classification accuracy for training in 86.7%, for validation 80%, misclassification was found belonging to grafted, stored at 2 °C. The detailed result of the model can be found in Table 6. The tendency of the separations of the LDA results can be seen in Figure 4. Root 1 (73.59%) mainly shows the separation between grafted and self-rooted types, while root 2 (19.33%) shows the separation between groups of fresh and stored melons. 

NIRS results for storage showed 100% classification for the different storage conditions of both grafted and self-root melons. For validation, melons stored at 2 °C had the highest accuracy irrespective of whether they were grafted or self-rooted. Self-rooted melons stored at 17 °C also showed a high validation accuracy. 

### 3.4. Results of Partial Least Square Regression Models for Sensory and Chemical Parameters Predicted from PLSR Results of the Electronic Tongue and NIRS

PLSR results predicted of the results of the electronic tongue and NIRS of the variety data set for chemical parameters and sensory parameters can be seen in Table 7. The best PLSR model for NIRS analysis showed an R^2^CV of 0.96 for the Brix° content and a low error of 0.29 g/mg. With the exception of β-carotene and total carotene, all the other standard analytical parameters had a low RMSECV between 0.15 g/mg–3.2 g/mg. In the case of the variety data set the best results were obtained for electronic tongue prediction of total carotene, β-carotene, cis-β-carotene, violaxanthin and vitamin C content, while in sensory parameters the taste persistency sweet aroma and juiciness of melons could be predicted with the highest correlation (R^2^C and R^2^CV > 0.8000) between the measured and predicted values. In the case of phytoene and phytofluene no good model was found.

In the case of the e-tongue analysis of storage data set (Table 8), the highest correlation between measured and predicted value was obtained in the case of ζ-carotene, Brix°, for sensory in the case of sweet taste and aftertaste. Prediction by the NIRS data set provided the best result for Brix° with the correlation of R^2^ = 0.9981 for recognition and R^2^CV = 0.9585 for validation. In the case of results of the sensory test, the aftertaste, melon aroma, sweet taste and taste persistency could be estimated with the highest correlation between measured and predicted values.

## 4. Discussion

### 4.1. Color Classification

Electronic tongue results showed complete separation of Cantaloupe type (orange-fleshed) and Galia type melons (green) from each other (Figure 3). This phenomenon was also observed in the results of standard analytical methods, where significant differences were found in all parameters, showing that these two types are different in chemical composition (Table 1). Discrimination between these two-colored melon types was also confirmed with the NIRS, where there was no misclassification of color. Discriminating melon colors is an important parameter that is often associated with ripeness and plays a key role in consumer perception of food items before purchase. Galia melon is a hybrid originating from a Cantaloupe–Honeydew cross; ripeness is measured not by softness at the stem, but rather by color and fragrance [32]. According to Sánchez et al. (2014), Cantaloupe type melons may be harvested, when the external color beneath the netting begins to change from green to yellow-green because the skin color gradually changes light yellow, but the orange-fleshed pulp requires non-destructive methods in order to avoid damage to the fruit at full ripeness [25]. With a non-permeable advantage, NIRS can be combined with existing methods for quick profiling of melons according to their color. Generally, the NIRS and E-tongue showed similar results for the discrimination of the two main types, results obtained in the detailed evaluation (Table 3, Table 4, Table 5, Table 6, Table 7 and Table 8). Were differences in the results of the two instruments, which may be due to their unique advantages and disadvantages showed in Table 9.

### 4.2. Melon Classification Based on Varieties

Sensory results of the Cantaloupe type melons showed that *Celestial* variety had the best result for most of the attributes investigated. This pattern was confirmed in the NIRS results in Table 5 where *Celestial* had the highest validation accuracy of 93.4%. *Centro* variety significantly differed from one of the other types in juiciness, texture, fermented aroma and fermented taste (Figure 2a), and this pattern was also distinguished in the LDA model built for the results of the electronic tongue from the *Donatello* and *Celestial* groups (Figure 3). 

Sensory results of Galia type melons shows significant difference only for fermented taste and aroma (Figure 2b); however, these type could be differentiated at high accuracy by the electronic tongue, showing that the electronic tongue is more sensible not only for differentiating different types of melons, but also different varieties in the same type. Classification accuracies from the NIRS were also in line with this observation and agreed with studies by Seregély et al. (2004) when they discriminated different varieties of hybrid melon with NIRS [33].

### 4.3. Melon Classification Based on Storage and Growth Conditions (Grafted or Self-Rooted)

Sensory results of storage data sets showed significant differences in eight parameters from the ten between grafted and self-root melons (Figure 2c,d), and these two types also could be distinguished from each other based on the results of the electronic tongue and NIRS (Table 6). The results obtained for classification of grafted and self-rooted melons with the electronic tongue are better than obtained for previous studies for watermelons, where technological and environmental conditions had a higher role in the differentiation of samples [21]. It is also true for the melons stored at 2 °C, showing that the electronic tongue is in accordance with the results of the sensory test (Table 6). Cantaloupe melons are recommended to be stored at temperatures of 0–5 °C for maximum preservation of their qualities [34]. The high classification accuracy (88.34% with an electronic tongue and 88.83% with NIRS) of self-rooted melons stored at a higher temperature (17 °C) compared to those stored at lower temperatures explains the influence of temperature on melon quality.

### 4.4. PLSR Prediction of Melon Qualities

In PLSR, an R^2^ close to 1 is a necessary condition for a good model [35], but this may not be the only requirement. The errors in calibration: root mean square error in cross validations (RMSECV) explains the fit of the observations to the model in both calibration and validation steps. It is a measure of the average difference between the values determined by the reference methods and those predicted by the model [36]. PLSR for electronic tongue provided the best results in the case of carotenes for the variety data set (Table 7), this could be the results of the significant differences between the varieties found for these parameters (Table 1). The same trend is true for the storage data set where Brix° provided the best results in PLSR (Table 8) and ANOVA test also found four different subsets for the five storage types (Table 2). With the exception of Brix°, the electronic tongue could generally, predict storage and standard analytical parameters better than the NIRS. Ref. [24] and Ref. [37] also reported much better results in their study for brix prediction in melon with NIRS. According to [33], the main difficulty of NIRS studies may be because NIRS measures the physical (optical) properties that are determined by chemical compounds and molecular structures, this may have accounted for the poor NIRS prediction results.

## 5. Conclusions

In the present study, we evaluated the influence of storage condition and grafting of melon cultivars testing and based on the instrumental and organoleptic examination, and according to the main question posed, we can conclude the follows:

ANOVA results showed significant differences in each standard analytical parameter between Cantaloupe type (orange-fleshed) and Galia type melons (green). There were also significant differences in the five melon varieties, according to their flesh color. Significant differences were also observed in different storage conditions. 

Correlative analytical techniques (electronic tongue and Metri NIRS) confirmed the ANOVA results with a complete separation of both Cantaloupe type (orange-fleshed) and Galia type melons (green), thus, ascertaining their chemical variations. *Celestial* variety had the best result for most of the attributes investigated, in agreement with the NIRS results. 

Sensory results of Galia type melons shows significant difference only for fermented taste and aroma; however, these typed could be differentiated at high accuracy by the electronic tongue. Sensory results of storage data sets showed significant differences in eight parameters from the ten between grafted and self-root melons, and these two types also could be distinguished from each other based on the results of the electronic tongue and NIRS. 

Partial least squares regression models with the electronic tongue provided the best accuracy in the case of carotenes for the variety data set and the best for brix in the storage data set. 

Generally, the electronic tongue could predict storage and standard analytical parameters better than the NIRS, but NIRS showed higher classification accuracies. Combining both electronic tongue and near-infrared spectroscopy provides a rapid, non-destructible means of monitoring melon varieties and the effect of storage on the quality of Cantaloupe type (orange-fleshed) and Galia type melons (green-fleshed). 

## Figures and Tables

**Figure 1 sensors-19-05010-f001:**
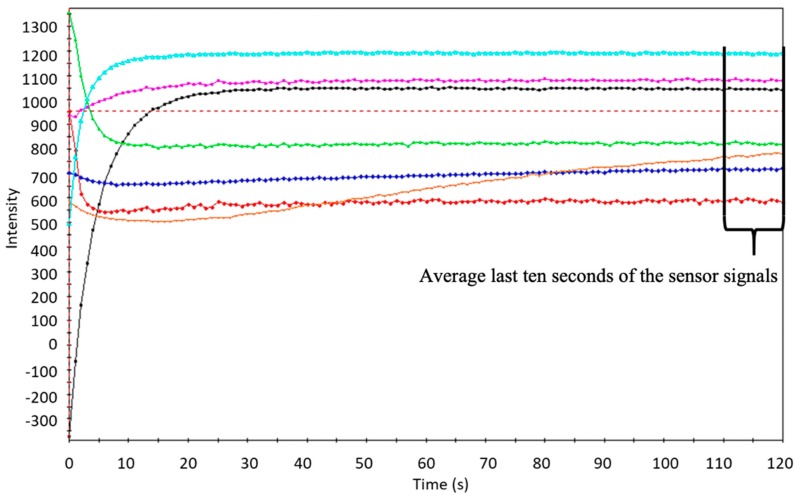
Electronic tongue sensor signals acquired in the 120 s of signal acquisition of one selected *Celestial* melon sample.

**Figure 2 sensors-19-05010-f002:**
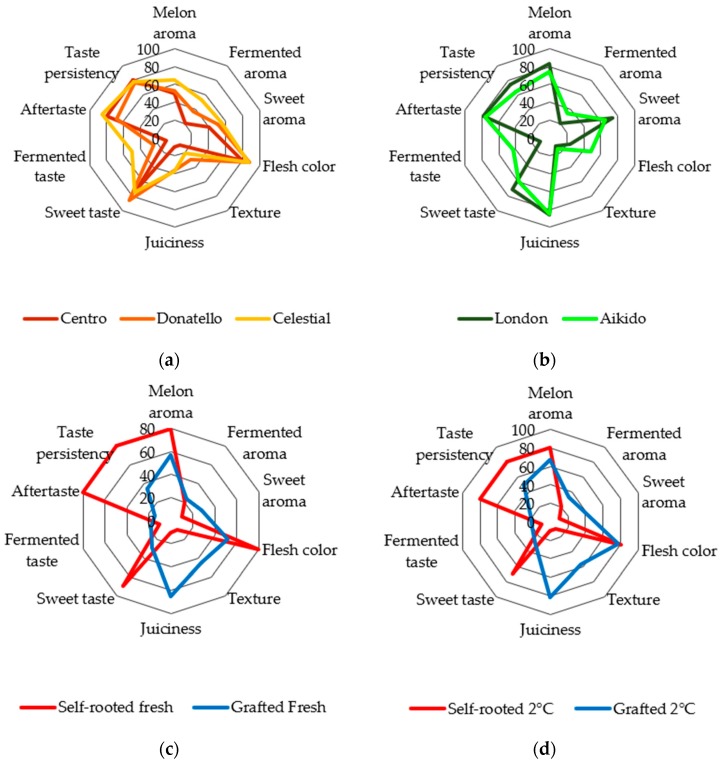
Results of the sensory profile test: (**a**,**b**) Results of the variety data set *n* = 11, (**c**,**d**) results of the storage data set *n* = 10.

**Figure 3 sensors-19-05010-f003:**
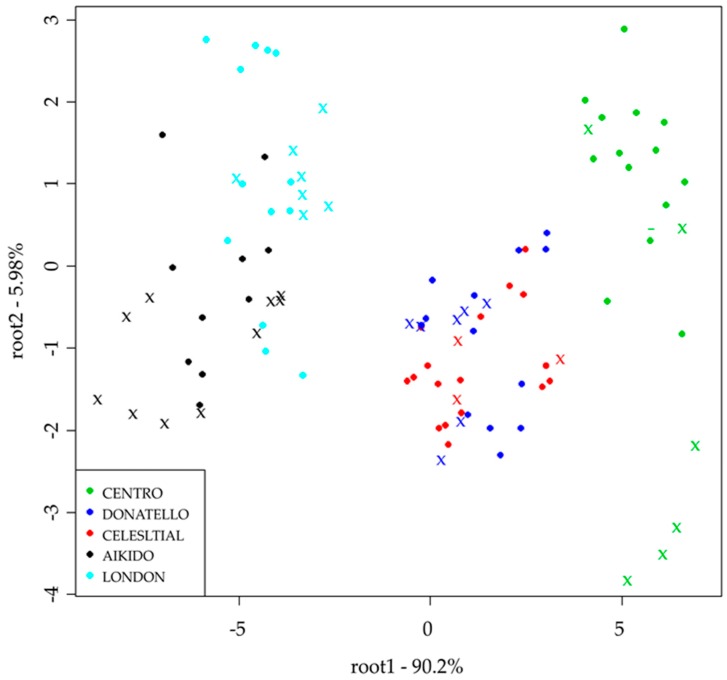
LDA classification results of the electronic tongue for differentiation of the five varieties after drift correction and outlier detection (*n* = 100) ●training ✖ validation.

**Figure 4 sensors-19-05010-f004:**
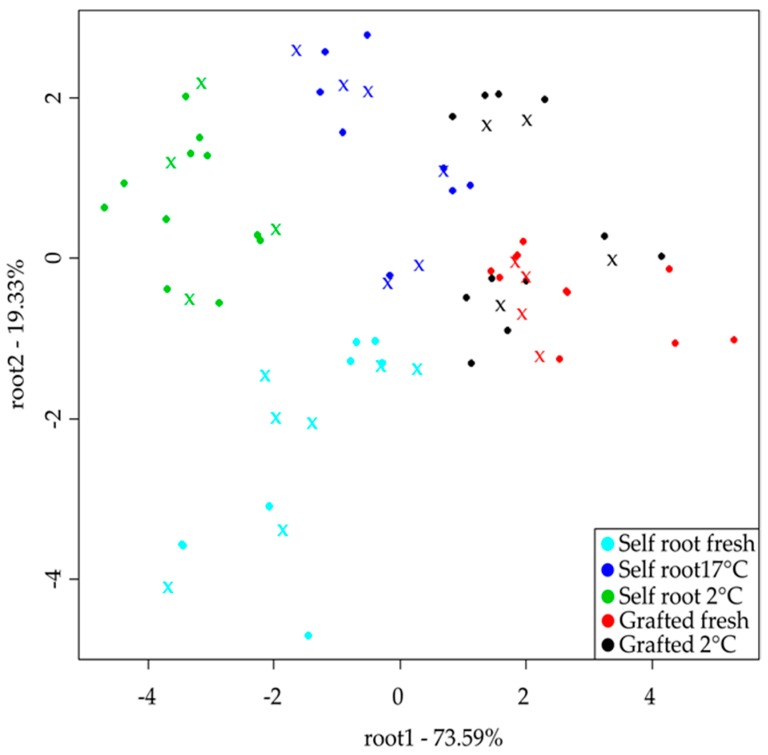
LDA classification results of the electronic tongue for differentiation of the five storage groups after drift correction and outlier detection (*n* = 77) ●training ✖ validation.

**Table 1 sensors-19-05010-t001:** Results of the standard analytical methods of the different varieties (values are in µg/mL).

	Galia Type		Cantaloupe Type		
	*Aikido*	*London*	*Celestial*	*Centro*	*Donatello*
**Phytoene**	-	-	3.545 ± 0.76^a^	2.919 ± 0.828^a^	2.367 ± 0.222^a^
**Phyto floene**	-	-	3.214 ± 0.731^a^	2.922 ± 0.758^a^	1.95 ± 0.314^a^
**Cis β-carotene**	0.011 ± 0^a^	0.085 ± 0.021^a^	0.539 ± 0.17^b^	0.76 ± 0.225^b^	0.441 ± 0.037^b^
**β-carotene**	0.601 ± 0.056^a^	3.606 ± 0.385^a^	48.074 ± 11.033^b^	66.056 ± 8.28^c^	31.895 ± 3.063^b^
**ζ-carotene**	0.018 ± 0.004^a^	0.085 ± 0.021^a^	3.079 ± 0.747^c^	2.674 ± 0.521^b,c^	1.791 ± 0.313^b^
**Mutatoxantin**	-	-	0.085 ± 0.021^a^	0.245 ± 0.139^a^	0.085 ± 0.021^a^
**Lutein**	0.822 ± 0.076^b^	0.871 ± 0.203^b^	0.417 ± 0.042^a^	0.502 ± 0.076^a^	0.282 ± 0.021^a^
**Violaxantin**	0.809 ± 0.037^d^	0.615 ± 0.053^c^	0.3 ± 0.028^a,b^	0.373 ± 0.092^b^	0.187 ± 0.031^a^
**Total carotene**	5.275 ± 1.658^a^	7.973 ± 1.888^a^	53.361 ± 11.408^b^	72.68 ± 8.28^c^	35.944 ± 4.598^b^
**Chlorophyll A**	2.122 ± 0.309^a^	1.705 ± 0.085^a^	-	-	-
**Chlorophyll B**	0.809 ± 0.133^b^	0.232 ± 0.021^a^	-	-	-
**Vitamin C**	-	0.15 ± 0.26^a^	34.917 ± 2.15^c^	26.317 ± 5.212^b^	30.74 ± 0.52^b,c^
**Brix°**	8.978 ± 1.357^a^	8.367 ± 0.633^a^	7.744 ± 0.707^a^	8.367 ± 3.221^a^	7.622 ± 0.662^a^

Mean ± Standard deviation, letters (a,b,c,d) are showing the significant differences between varieties per each parameter resulted by ANOVA test followed by Tukey-HSD post hoc test at *p* < 0.05.

**Table 2 sensors-19-05010-t002:** Results of standard analytical methods for the storage test set (values are in µg/mL).

	Grafted Fresh	Grafted 2 °C	Self-Rooted Fresh	Self-Rooted 2 °C	Self-Rooted 17 °C
**Phytoene**	1.68 ± 0.62^a^	1.13 ± 0.33^a^	2.98 ± 0.63^b^	1.66 ± 0.33^a^	1.79 ± 0.4^a,b^
**Phytofluene**	2.09 ± 0.87^a,b^	0.95 ± 0.18^a^	3.56 ± 0.85^b^	1.46 ± 0.49^a^	1.73 ± 0.23^a^
**Cis β-Carotene**	3.8 ± 2.22^a^	1.07 ± 0.64^a^	10.43 ± 2.76^b^	1.05 ± 0.35^a^	1.25 ± 0.41^a^
**β-Carotene**	71.39 ± 18.77^b^	30.91 ± 3.68^a^	116.41 ± 16.74^c^	48.21 ± 13.98^a,b^	54.49 ± 9.6^a,b^
**ζ-Carotene**	3.32 ± 0.54^a^	1.5 ± 0.17^a^	7.48 ± 2.22^b^	2.63 ± 0.81^a^	3.41 ± 0.63^a^
**Mutatoxanthin**	0.28 ± 0.09^b^	0.13 ± 0.08^a,b^	0.23 ± 0.08^a,b^	0.06 ± 0.01^a^	0.13 ± 0.02^a,b^
**Lutein**	0.5 ± 0.09^b^	0.24 ± 0.02^a^	0.54 ± 0.11^b^	0.38 ± 0.13^a,b^	0.24 ± 0.04^a^
**Violaxanthin**	0.07 ± 0.04^a^	0.17 ± 0.02^a,b^	0.28 ± 0.15^a,b^	0.32 ± 0.08^b^	0.27 ± 0.06^a,b^
**Total carotene**	81.14 ± 20.06^b^	35.64 ± 2.89^a^	129.72 ± 12.7^c^	54.14 ± 13.19^a,b^	61.27 ± 8.28^a,b^
**Brix°**	5.67 ± 0.32^b^	4.41 ± 0.32^a^	8.57 ± 1^d^	7.56 ± 1.16^c,d^	7.21 ± 0.63^c^
**Vitamin C**	-	132.29 ± 27.78^c^	-	5.49 ± 2.16^b^	1.93 ± 0.84^a^

Mean ± Standard deviation, letters are showing the significant differences between varieties per each parameter resulted by ANOVA test followed by Tukey-HSD post hoc test at *p* < 0.05.

**Table 3 sensors-19-05010-t003:** Linear discriminant analysis (LDA) classification results of the variety data set based on the results of electronic tongue (ET) and near-infrared spectroscopy (NIRS).

		Electronic Tongue							NIRS					
Color	Total Accuracy	Varieties	*Celestial*	*Centro*	*Donatello*	*Aikido*	*London*	Total Accuracy	Varieties	*Celestial*	*Centro*	*Donatello*	*Aikido*	*London*
Yellow	Recognition 85.51%	*Celestial*	72.49	0	24.98	0	0	Recognition 100%	*Celestial*	100	0	0	0	0
Yellow		*Centro*	0	100	0	0	0		*Centro*	0	100	0	0	0
Yellow		*Donatello*	27.51	0	75.02	0	0		*Donatello*	0	0	100	0	0
Green		*Aikido*	0	0	0	90.02	9.98		*Aikido*	0	0	0	100	0
Green		*London*	0	0	0	9.98	90.02		*London*	0	0	0	0	100
Yellow	Cross validated 59.03%	*Celestial*	25.04	0	74.96	0	0	Cross Validated 54.75%	*Celestial*	80.16	0	0	6.6	6.61
Yellow		*Centro*	0	95.05	0	0	0		*Centro*	0	46.69	0	20	6.61
Yellow		*Donatello*	74.96	4.95	25.04	0	0		*Donatello*	6.61	20.04	73.4	40	6.61
Green		*Aikido*	0	0	0	80.03	29.99		*Aikido*	6.61	6.61	26.6	20	26.65
Green		*London*	0	0	0	19.97	70.01		*London*	6.61	26.65	0	13.4	53.51

**Table 4 sensors-19-05010-t004:** LDA classification results of Galia type melons based on the results of ET and NIR.

	Electronic Tongue			NIRS			
Total Accuracy	Varieties	*Aikido*	*London*	Total Accuracy	Varieties	*Aikido*	*London*
Recognition 89.99%	*Aikido*	92.5	12.52	Recognition 100%	*Aikido*	100	0
	*London*	7.5	87.48		*London*	0	100
Cross Validated 87.49%	*Aikido*	89.96	14.99	Cross Validated 90%	*Aikido*	100	20
	*London*	10.04	85.01		*London*	0	80

**Table 5 sensors-19-05010-t005:** LDA classification results of Cantaloupe type melons based on results of ET and NIR.

	Electronic Tongue				NIRS				
Total Accuracy	Varieties	*Celestial*	*Centro*	*Donatello*	Total Accuracy	Varieties	*Celestial*	*Centro*	*Donatello*
Recognition 82.33%	*Celestial*	77.75	0	30.77	Recognition 100%	*Celestial*	100	0	0
	*Centro*	0	100	0		*Centro*	0	100	0
	*Donatello*	22.25	0	69.23		*Donatello*	0	0	100
Cross Validated 51.19%	*Celestial*	25	0	71.43	Cross Validated 64.47%	*Celestial*	93.4	0	26.6
	*Centro*	0	100	0		*Centro*	6.6	66.6	40
	*Donatello*	75	0	28.57		*Donatello*	0	33.4	33.4

**Table 6 sensors-19-05010-t006:** LDA classification results of the storage data set based on the results of ET and NIRS.

		Electronic Tongue					NIR						
Accuracy	Storage Type	Grafted Fresh	Grafted 2 °C	Self Root Fresh	Self Root 2 °C	Self Root 17 °C	Accuracy	Storage type	Grafted Fresh	Grafted 2 °C	Self Root fresh	Self Root 2 °C	Self Root 17 °C
Training 92.10%	Grafted fresh	86.7	20	0	0	2.91	Training 100%	Grafted fresh	100	0	0	0	0
	Grafted 2 °C	13.3	76.7	0	0	0		Grafted 2 °C	0	100	0	0	0
	Self root fresh	0	0	100	0	0		Self root fresh	0	0	100	0	0
	Self root 2 °C	0	0	0	100	0		Self root 2 °C	0	0	0	100	0
	Self root 17 °C	0	3.3	0	0	97.09		Self root 17 °C	0	0	0	0	100
Validation 87.03%	Grafted fresh	80	13.4	0	0	5.83	Validation 84.46%	Grafted fresh	77.83	5.51	0	0	0
	Grafted 2 °C	20	80	6.6	0	5.83		Grafted 2 °C	11.17	88.98	5.5	0	0
	Self root fresh	0	0	93.4	6.6	0		Self root fresh	5.5	5.51	77.83	11.17	0
	Self root 2 °C	0	0	0	93.4	0		Self root 2 °C	0	0	5.5	88.83	11.17
	Self root 17 °C	0	6.6	0	0	88.34		Self root 17 °C	5.5	0	11.17	0	88.83

**Table 7 sensors-19-05010-t007:** Results of PLSR for the standard analytical parameters (values are in µg/mL) and sensory parameters predicted by ET and NIRS for variety data set.

		Electronic Tongue						NIRS					
	Parameters	Latent variables	Data Points	R^2^C	RMSEC	R^2^CV	RMSECV	Latent Variables	Data Points	R^2^C	RMSEC	R^2^CV	RMSECV
Standard analytical properties	**Brix°**	5	56	0.4238	0.9949	0.2633	1.1241	20	45	0.9981	0.0625	**0.9585***	0.2914
	**β-carotene**	6	56	0.89	7.3223	0.8516	8.4964	17	45	0.9462	5.9835	0.4645	18.87
	**Cis-β-carotene**	6	56	0.8987	0.0915	0.8672	0.1046	10	45	0.8279	0.1244	0.3683	0.2384
	**Chlorophyll A**	2	24	0.7432	0.1543	0.6330	0.1840	3	18	0.3738	0.2204	0.0831	0.2666
	**Chlorophyll B**	2	24	0.6978	0.1829	0.5639	0.2194	4	18	0.6263	0.1826	0.1345	0.278
	**Luthein**	4	56	0.6651	0.1503	0.5821	0.1677	10	45	0.7611	0.12	0.2719	0.2095
	**Total carotene**	6	56	0.8914	7.5094	0.8528	8.7341	17	45	0.9468	6.1248	0.461	19.487
	**Violaxanthin**	5	56	0.8108	0.1068	0.7632	0.1194	12	45	0.8868	0.0771	0.5461	0.1545
	**Vitamin C**	6	55	0.8967	5.2308	0.8653	5.9682	20	27	0.9993	0.1154	0.4456	3.284
	**ζ-carotene**	5	56	0.7935	0.5842	0.7402	0.6547	11	45	0.8503	0.5123	0.4768	0.9578
Sensory properties	**Aftertaste**	1	100	0.0354	5.4113	0.0112	5.4783	5	65	0.29	4.7568	0.0564	5.4839
	**Flesh color**	5	100	0.6976	13.939	0.6494	15.0047	3	64	0.2617	21.941	0.1917	22.959
	**Fermented taste**	4	100	0.3362	13.4063	0.2486	14.2658	5	63	0.4427	12.277	0.2648	14.101
	**Fermented aroma**	5	100	0.2732	10.3195	0.1739	10.9991	5	67	0.2828	10.208	0.0486	11.757
	**Melon aroma**	4	100	0.7324	6.5185	0.7007	6.8916	5	63	0.5666	8.2204	0.349	10.074
	**Sweet aroma**	4	100	0.8186	4.9681	0.7962	5.2654	5	61	0.6613	6.7504	0.572	7.5884
	**Sweet taste**	5	100	0.3231	6.9736	0.2277	7.4468	5	67	0.4238	6.6587	0.2651	7.5201
	**Taste persistency**	5	100	0.803	2.4503	0.7761	2.612	5	59	0.7694	2.634	0.6815	3.0959
	**Texture**	5	100	0.2017	6.5978	0.0995	7.0073	4	60	0.0631	7.0443	0.0043	7.2623
	**Juiciness**	5	100	0.9206	8.4033	0.9087	9.0062	4	60	0.6787	16.415	0.6164	17.936

R^2^C: Coefficient of determination; RMSEC: Root meant error of calibration; R^2^CV: Coefficient of determination of cross validation; RMSECV; Root mean square error cross validation.

**Table 8 sensors-19-05010-t008:** Results of PLSR for the standard analytical and sensory parameters predicted by ET and NIRS for the storage data set.

		Electronic Tongue						NIRS					
	Parameters	Latent Variables	Data Points	R^2^C	RMSEC	R^2^CV	RMSECV	Latent Variables	Data Points	R^2^C	RMSEC	R^2^CV	RMSECV
Standard analytical properties	**Brix°**	1	50	0.816	0.6806	0.8002	0.7088	12	26	0.9992	0.0459	0.8144	0.6827
	**β-carotene**	4	44	0.4975	22.1737	0.3256	25.6006	15	31	0.9424	8.5495	−0.012	35.83
	**Cis-β-carotene**	4	44	0.6083	2.4284	0.4579	2.8537	15	32	0.9009	1.3535	−0.7949	5.7594
	**Lutein**	2	44	0.4167	0.1119	0.2942	0.123	6	34	0.6707	0.0789	0.3939	0.1071
	**Mutatoxanthin**	4	44	0.5812	0.0611	0.3668	0.075	15	35	0.9488	0.0197	−0.7132	0.1138
	**Phytoene**	3	44	0.5141	0.509	0.3423	0.5912	15	32	0.9233	0.2114	−1.0245	1.0861
	**Phytofluene**	3	44	0.4029	0.7909	0.2069	0.9103	15	31	0.8681	0.3804	−2.5857	1.9833
	**Total carotene**	4	44	0.5446	22.9254	0.3812	26.6193	15	31	0.9717	6.3529	0.3484	30.486
	**Violaxanthin**	2	44	0.3022	0.092	0.1165	0.1034	15	30	0.9653	0.0209	0.0226	0.111
	**Vitamin C**	2	26	0.6897	34.9167	0.6054	39.3004	15	25	0.9684	9.3388	−1.5506	83.949
	**ζ-carotene**	4	44	0.7914	1.0283	0.6613	1.3087	15	32	0.8962	0.675	−5.037	5.1466
Sensory properties	**Aftertaste**	1	60	0.7987	13.8018	0.7849	14.2621	10	57	0.9465	7.0339	0.8525	11.68
	**Flesh color**	2	60	0.4079	9.2906	0.3198	9.9535	10	56	0.8184	5.2774	−0.0275	12.552
	**Fermented taste**	1	60	0.7665	2.2761	0.7496	2.3565	10	61	0.906	1.4383	0.7737	2.2312
	**Fermented aroma**	1	60	0.4747	3.9534	0.4355	4.0971	10	56	0.8635	1.9815	0.6549	3.1509
	**Melon aroma**	2	60	0.7297	5.0735	0.6945	5.3917	10	54	0.9605	1.9787	0.8641	3.672
	**Sweet aroma**	3	60	0.7556	6.0293	0.7136	6.5138	10	61	0.8868	4.0793	0.7286	6.3175
	**Sweet taste**	1	60	0.8038	9.1659	0.7901	9.4765	10	59	0.9458	4.7997	0.8503	7.975
	**Taste persistency**	2	60	0.772	9.4545	0.7433	10.028	10	54	0.9438	4.6644	0.8334	8.0325
	**Texture**	1	60	0.7651	10.1466	0.748	10.5051	10	61	0.9056	6.4052	0.7737	9.9156
	**Juiciness**	1	60	0.7738	15.3924	0.7574	15.9337	10	61	0.9036	10.009	0.745	16.279

**Table 9 sensors-19-05010-t009:** Properties of the electronic tongue and NIRS as analytical methods.

Attribute/Function	E-Tongue	NIRS
Quick	Yes	Yes
Non-destructive	No	Yes
No waste (use of reagents)	Yes	Yes
Less labor	Yes	Yes
Relatively low cost and safe application	Yes	Yes
Sophisticated	No	Yes
Drift	Yes	No
High selectivity and sensitivity	Yes	Yes
Small sample size required	No	Yes
Quantification and classification	Yes	Yes
Temperature sensitive	Yes	Yes
Advanced (complex) data analysis	Yes	Yes
Small and portable instrument size	No*	Yes
High precision	Yes	Yes
Calibration is dependent on food constituent	Yes	Yes
Easy to install (maintenance)	Yes	Yes
Flexibility (simultaneous analysis)	Yes	Yes
Dependence on reference information	Yes	Yes

* commercial E-tongue instruments are usually bench-top types.

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
