# Peer review of "Standard Analytical Methods, Sensory Evaluation, NIRS and Electronic Tongue for Sensing Taste Attributes of Different Melon Varieties"

_sensors, 2019, doi:10.3390/s19225010_

Round 1

Reviewer 1 Report

The work is correct, contains good measurement methodology and the results obtained are correctly interpreted. However, I miss the scientific news here and why it was made. I hope that in the improved version the authors will explain it better. My other comments are below:

Introduction - There is no clear scientific purpose. What is new in this research? The authors must demonstrate this.

2.3. Electronic tongue measurements - the following sentence is quite illegible: "Each sequence was measured in four rounds, resulting in 20 consecutive measurement for all samples" .Does it mean that only 20 tests were performed? It is probably not enough to generalize obtained information from research. Please extend this issue.

Discussion - it would be worth inserting a table with the pros and cons of ET and NIRS methods for classification under various characteristics. This would make it easier for the reader to track all these tables.

Author Response

Please see the response in the attachment.

Reviewer 2 Report

The authors used HPLC measurement, classical sensory evaluation, electronic tongue and NIR to classify the different melon varieties and different storage conditions. The study is quite throughout, and the detailed results are presented in the manuscript. There’re only a few minor issues I’ve noticed in the manuscript, as listed below:

The authors did not show any measurements plot using electronic tongues and NIR, instead they directly showed Table 3 to Table 6, where each variety of the melons (or storage conditions) are being compared and discriminated. It seems to be a bit abrupt when I read Tables 3 to 6. Would it be better to show one plot of the results of different melons tested by the electronic tongues, and one plot for NIR? This way the readers will have a better understanding of how Tables 3 to 6 are obtained. In the experimental section, what is the measurement technique for electronic tongues? Is it cyclic voltammetry? This is one reason I wish there are some plots of the actual measurement. There’re a few typo or format issues. In line 50, it should be ‘0.75 – 35.3’ instead of ‘0,75 – 35,3’. The a,b,c in Table 1 should be superscript I guess, considering this is how it looks in Table 2. Also although the authors kept talking about vitamin C, in the tables it’s shown as ascorbic acid. Maybe the authors should make it more consistent. There’re also a few locations where the reference source can not be found.

Author Response

(The authors gave the same response as above.)

Round 2

Reviewer 1 Report

The changes introduced by the authors are correct in my opinion. In particular, Table 9 simply compares the pros and cons of the measurement methods used. I believe that work deserves further stages of evaluation and I recommend it.